# Evaluation and Comparison of Semantic Segmentation Networks for Rice Identification Based on Sentinel-2 Imagery

**Huiyao Xu** [1,2]**, Jia Song** [1,3],*[ID] **and Yunqiang Zhu** [1]

1  State Key Laboratory of Resources and Environmental Information System, Institute of Geographic Sciences and Natural Resources, Chinese Academy of Sciences, Beijing 100101, China
2  School of Resources and Environment, University of Chinese Academy of Sciences, Beijing 100049, China
3  Jiangsu Center for Collaborative Innovation in Geographical Information Resource Development and Application, Nanjing 210023, China
*  Correspondence: songj@igsnrr.ac.cn

**Abstract:** Efficient and accurate rice identification based on high spatial and temporal resolution remote sensing imagery is essential for achieving precision agriculture and ensuring food security. Semantic segmentation networks in deep learning are an effective solution for crop identification, and they are mainly based on two architectures: the commonly used convolutional neural network (CNN) architecture and the novel Vision Transformer architecture. Research on crop identification from remote sensing imagery using Vision Transformer has only emerged in recent times, mostly in sub-meter resolution or even higher resolution imagery. Sub-meter resolution images are not suitable for large scale crop identification as they are difficult to obtain. Therefore, studying and analyzing the differences between Vision Transformer and CNN in crop identification in the meter resolution images can validate the generalizability of Vision Transformer and provide new ideas for model selection in crop identification research at large scale. This paper compares the performance of two representative CNN networks (U-Net and DeepLab v3) and a novel Vision Transformer network (Swin Transformer) on rice identification in Sentinel-2 of 10 m resolution. The results show that the three networks have different characteristics: (1) Swin Transformer has the highest rice identification accuracy and good farmland boundary segmentation ability. Although Swin Transformer has the largest number of model parameters, the training time is shorter than DeepLab v3, indicating that Swin Transformer has good computational efficiency. (2) DeepLab v3 also has good accuracy in rice identification. However, the boundaries of the rice fields identified by DeepLab v3 tend to shift towards the upper left corner. (3) U-Net takes the shortest time for both training and prediction and is able to segment the farmland boundaries accurately for correctly identified rice fields. However, U-Net's accuracy of rice identification is lowest, and rice is easily confused with soybean, corn, sweet potato and cotton in the prediction. The results reveal that the Vision Transformer network has great potential for identifying crops at the country or even global scale.

**Keywords:** rice identification; semantic segmentation; Swin Transformer; DeepLab v3; U-Net

## 1. Introduction

Rice is the staple food for more than half of the world's population and accounts for more than 12% of the world's farmland [1]. Rapid and precise identification of rice can be used to accurately assess rice yields, which is important for improving the efficiency of arable land use and achieving sustainable food production [2–5]. With the development of Earth observation technology, high spatial and temporal resolution satellites provide a more accurate and large-scale data source for exploring the spatial distribution of rice at a low cost [6]. Especially, the images from optical satellites facilitate the analysis of the spectral characteristics of crops due to their multiband nature [7–11].

With the high spatial and temporal resolution satellite images, various methods have been used for rice or crop identification [12], including the threshold-based method [13,14],

swallow machine learning method [15,16], and deep-learning-based method [17–20]. The semantic segmentation networks in deep learning have shown unprecedented performance when compared to other methods since much more and complex spectral features can be automatically learned for crop identification [21–27]. Until the emergence of the self-attention-based Vision Transformer networks in recent years, the convolutional neural networks (CNNs) have been the dominant architecture of the semantic segmentation for many years [28–31]. The CNNs can efficiently learn multi-dimensional features from large numbers of images through shared weights and locally aware convolutional operations, making them ideal for extracting spatial information and rich spectral features from remote sensing images [32–39]. U-Net [40] and DeepLab v3 [41] are two representative CNNs for crop identification, both of which can extract multi-scale features and are the baseline networks for many crop identification networks. U-Net [42] is heavily used in crop identification with large quantities of data due to its simple structure and a small number of layers. DeepLab v3 [43] is very sensitive to multi-scale information and can identify crops at different scales in complex contexts.

Since 2020, the novel Vision Transformer networks have drawn great attention in the computer vision field. Unlike CNNs, the Vision Transformer networks are built on a self-attention mechanism and aims to find out which regions of an image should be focused [44]. Because Transformer is computed in parallel for each pixel, both local and global information are well obtained. The Vision Transformer networks also have a larger field of perception than CNNs and are able to obtain richer global information. The novel Vision Transformer also began to be used for cropland extraction [45] and to be coupled with CNNs for crop identification [46] recently. The study [45] uses single-temporal Google Maps images and the DeepGlobe dataset, and proposes a boundary enhancement segmentation network based on Swin Transformer to enhance the boundary details of cropland. The study [46] also uses single-temporal sub-meter RGB images from Unmanned Aerial Vehicles (UAVs) for crop segmentation based on a coupled CNN and Transformer network (CCTNet), and shows that the CCTNet outperforms CNNs and Transformer networks. It can be seen that the existing crop or cropland identification studies have not compared the performance of CNNs and Transformers based on multi-temporal images, and the time-varying is a very important feature to distinguish different types of crops, or crops and non-crops [47–49]. In addition, the study [45,46] uses sub-meter remote sensing images for cropland and crop identification, which are suitable for crop or cropland mapping at the small-region scale. Thus, the performance comparison of CNNs and Transformers for crop identification at the global or country scale has not been specifically investigated.

Therefore, this study comprehensively compares two representative CNN-based networks: U-Net, DeepLab v3, and a novel Swin Transformer network for rice identification. The aspects of comparison include accuracy, running time, farmland boundary segmentation details, and the main crop type misclassified as rice. Multi-temporal Sentinel-2 images based on three key times of the rice growth cycle were used for comparing and analyzing the above three networks. The 10 m Sentinel-2 images are suitable for crop mapping with high spatio-temporal precision at the country or even global scale [50].

The remainder of this paper is structured as follows. Section 2 describes the study area we focus on and the data sources we use. Section 3 describes the general experimental structure and the evaluation metrics used in this paper. Section 4 reports the results of the experiments on rice identification. Section 5 provides a comprehensive discussion of the experimental results. Section 6 concludes this paper and discusses future works.

## 2. Materials

### 2.1. Study Area

The study area is located in Arkansas, USA (33°00′16′′–36°29′58′′N, 89°38′39′′–94°37′05′′W), as shown in Figure 1a,b. The region is located in the middle and lower Mississippi River and has a temperate climate with a long summer and short winter, an average annual rainfall of 1220 mm, and abundant water resources. Arkansas

is the number one producer of rice in the United States, accounting for over 40% of the country's rice production [51]. The eastern Mississippi River Valley is Arkansas' main crop growing area, with rice, cotton, soybeans, corn, and peanuts as the main crops, as shown in Figure 1b,c. The large amount of rice grown and the abundance of crop types in Arkansas enables the testing of the models' ability to identify rice in complex environments and facilitates analysis of crop types that are easily misclassified as rice.

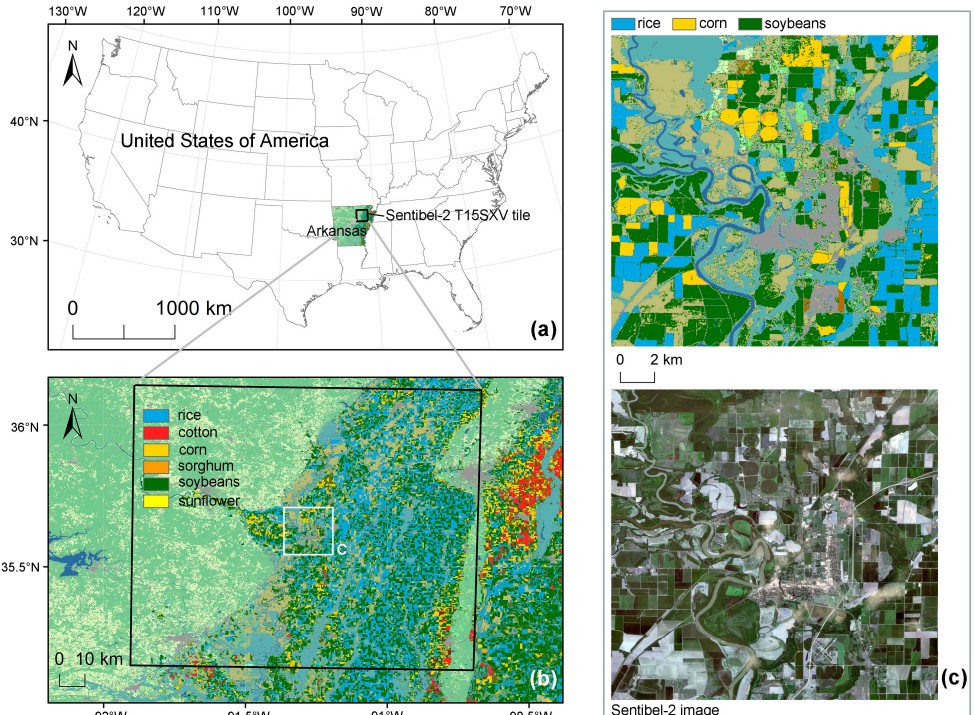

**Figure 1.** Study area overview. (**a**) Location of the experimental site. (**b**) Main crop type of the study area. (**c**) Details of the study area.

## 2.2. Datasets

### 2.2.1. Sentinel-2 and Preprocessing

Sentinel-2 imagery (downloaded at https://scihub.copernicus.eu/, accessed on 17 March 2022) is chosen as the data source for this study, with a spatial resolution of 10 m and a temporal resolution of 5 days. The major fertility periods for rice in Arkansas are concentrated in April–September, including the seedling, tillering, heading, and maturity stages. Therefore, the images are acquired from April to September 2020. To ensure the availability of the images, 27 scenes of Sentinel-2 imagery with cloudiness within 20% are selected as the research data. The preprocessing steps for the Sentinel-2 data are shown below:

(1) NDVI calculation. NDVI maps are calculated for each image at each time point.

(2) Maximum composite. Based on the maximum composite method, one NDVI map is synthesized every two months, corresponding to the early (April and May), middle (June and July), and late (August and September) rice growth periods, and the three NDVI synthesis maps are integrated into a single bimonthly NDVI map. The NDVI values in the cloud-covered areas are relatively low, so the maximum composite can effectively reduce the influence of clouds on rice identification [52,53].

Figure 2 shows the change in the NDVI of rice in the study area from April to September 2020. It can be seen that rice growth can be divided into three stages: (1) the stable low-value period (April–May), when rice is not sown or is in the early seedling stage, and NDVI values are low and stable; (2) the rapidly rising period (June–July), when rice is in a period of rapid growth from the tillering stage to the heading stage, with NDVI values rising rapidly until they reach a peak; and (3) the declining period (August–September),

during which rice continues to grow until maturity, and NDVI values begin to decline after reaching their peak. For these three stages of NDVI change, an NDVI image corresponding to the different periods is synthesized every two months in this paper.

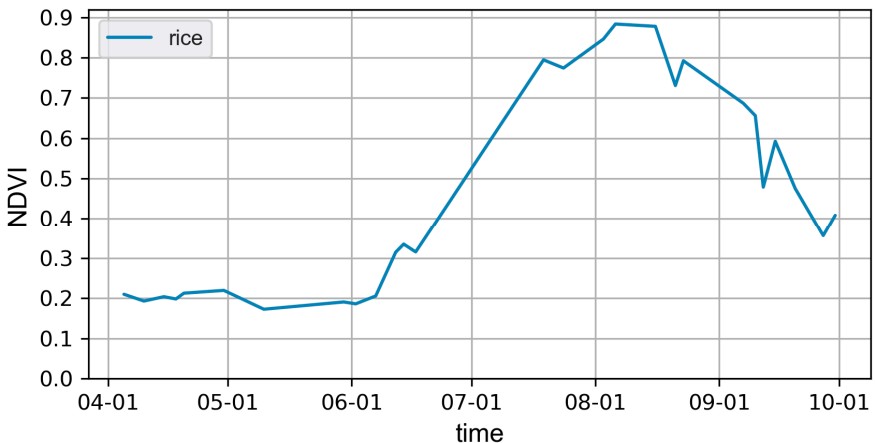

**Figure 2.** NDVI of rice in the study area between April and September 2020.

(3) The bimonthly NDVI map is cut into 100 images of 1098 × 1098 pixels, and 58 of them are selected as research sites in areas with a large amount of rice cultivation, 48 of which are used as training data and 10 as test data.

### 2.2.2. Reference Data and Preprocessing

The Arkansas 30 m Cropland Data Layer (CDL) published annually by the United States Department of Agriculture (USDA) (download at https://nassgeodata.gmu.edu/CropScape/, accessed on 11 January 2021.) is selected as reference data. These data are based on Landsat 8 OLI/TIRS, the Disaster Monitoring Constellation (DMC) DEIMOS-1, LISS-3, and Sentinel-2 satellites from 1 October 2019–31 December 2020. Crop-type data are obtained from a comprehensive field survey. The producer's precision and user's precision for the rice samples are 91.9% and 97.3%, respectively. The preprocessing steps for CDL data are shown in Figure 3.

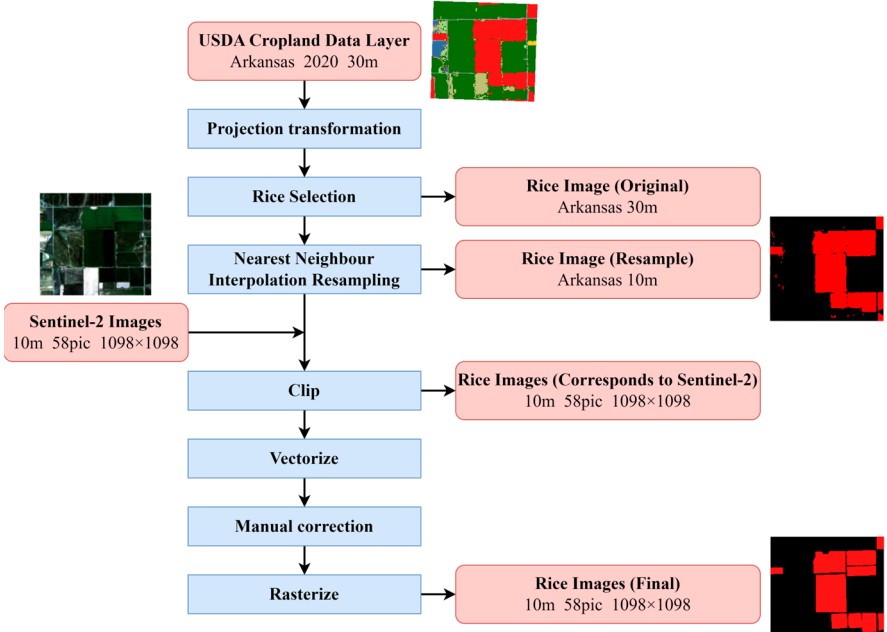

**Figure 3.** Reference data preprocessing steps.

(1) Projection transformation. Conversion of CDL data from the Albers projection to the UTM projection corresponding to Sentinel-2 imagery.

(2) 10 m rice map production. Selecting rice pixels from the projection-transformed CDL data and resampling the 30 m rice map to 10 m by nearest-neighbor interpolation.

(3) Sample production. The Arkansas rice map is cropped according to the 58 bi-monthly NDVI data mentioned above, and 58 1098 × 1098 pixel rice samples are obtained by manually correcting the field boundaries and manual denoising processes.

## 3. Methodology

We use three typical semantic segmentation networks (U-Net [54], DeepLab v3 [55] and Swin Transformer [56]) to build rice identification models. The evaluation method of them is shown in Figure 4. Fifty-eight sets of 1098 × 1098 rice samples are prepared from preprocessed Sentinel-2 and USDA CDL data, of which 48 are used for model training and 10 for testing. For the prediction results, we calculate the rice identification accuracy and the running efficiency of the three models. We also provide further specific analysis of the rice maps predicted by each model, including an analysis of the differences between the predicted results and the reference data and an analysis of the rice segmentation details. Additionally, we count the pixels for the categories that are incorrectly predicted and analyze crops that are easily confused with rice in crop classification.

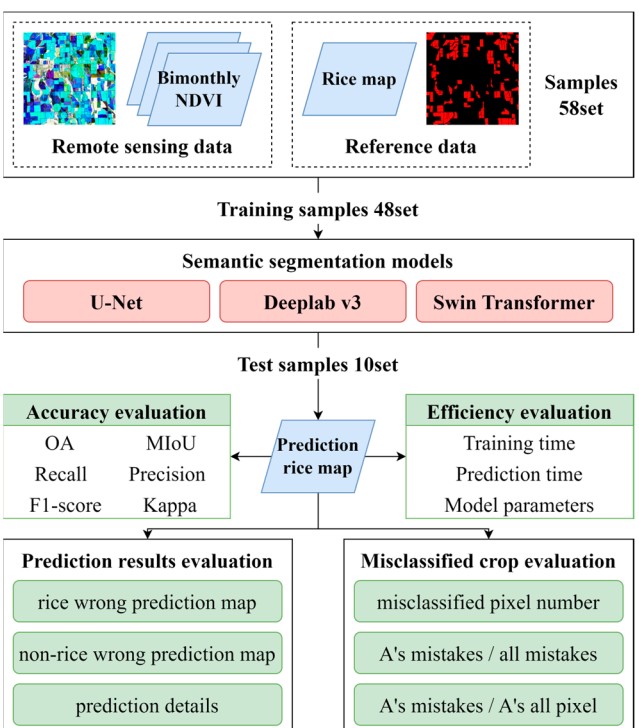

**Figure 4.** Evaluation framework.

### 3.1. Three Semantic Segmentation Networks for Rice Identification

U-Net [54], DeepLab v3 [55] and Swin Transformer [56] are three baseline networks with different structures. They have developed a variety of variants with different effects in the semantic segmentation task, all of which are based on the core structure of these three networks. To compare the effectiveness of these three networks for rice identification in high-resolution remote sensing images, we built the primitive U-Net, DeepLab v3 and Swin Transformer and conducted a comparative experiment of Sentinel-2-based rice identification models in Arkansas.

The core structure of U-Net is its symmetric encoder–decoder structure, skip connection and transpose convolutional upsampling approach, as shown in Figure 5. The U-Net's

encoder is the feature extraction path, which is responsible for extracting the spatial and temporal features from the three bimonthly NDVI data. The decoder is the image resolution recovery path, which uses the transpose convolution operation to stage a recovery of the image resolution. In addition, skip connection operation is used to connect the feature maps of the encoder and decoder. The skip connection cleverly fuses the low-level features and high-level features of the network, allowing the network to learn both low-level rice morphological features and high-level complex semantic features.

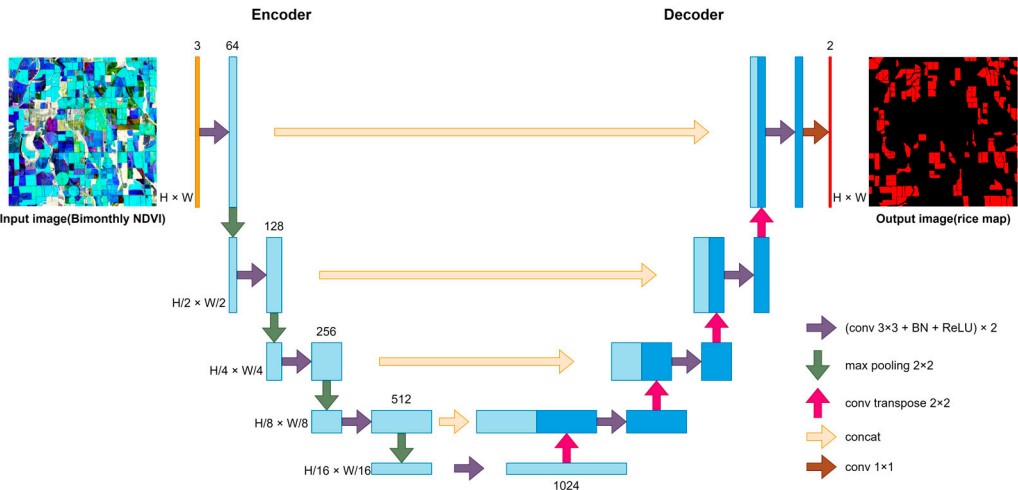

**Figure 5.** U-Net for rice identification [54].

The DeepLab v3 is composed of a backbone network for extracting feature and ASPP (atrous spatial pyramid pooling) module for classification. We uses ResNet-50 [57] as the backbone of DeepLab v3 for effectively mitigating the problem of network degradation. The ASPP is a parallel atrous convolution structure, as shown in Figure 6. The atrous convolution structure can expand the network perceptual field without reducing the image resolution. Larger perceptual field is helpful to capture larger context information, thereby distinguishing rice from other objects. In addition, the ASPP captures multi-scale information by using multiple different atrous rates and thus can identify rice based on the multi-scale features in the image.

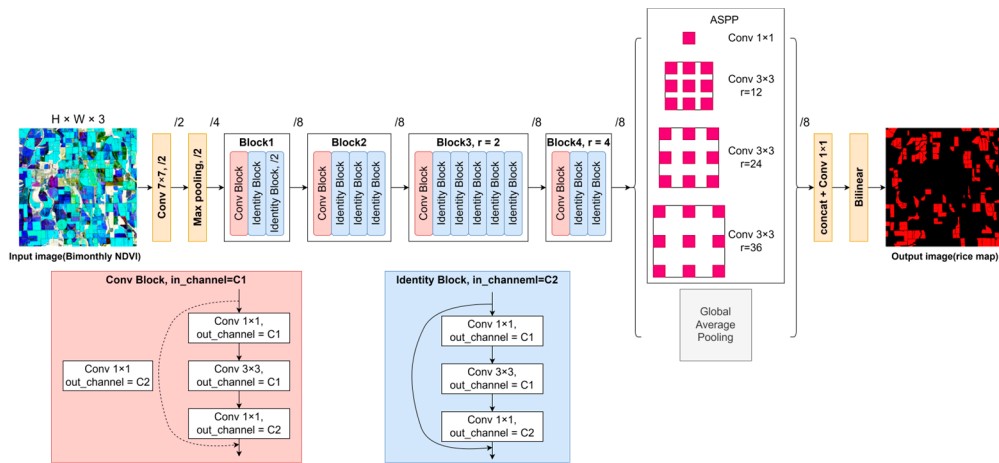

**Figure 6.** DeepLab v3 (backbone is ResNet50) for rice identification [55].

The Swin Transformer is a network entirely based on the attention mechanism and shifted windows, as shown in Figure 7. The attention score for each pixel is obtained by computing the relationship between each pixel and all remaining pixels in the window. It helps to find the more rice-concerned features from the images. Moreover, the Swin

Transformer has a hierarchical structure similar to the CNN network architecture, reducing the image size in stages while increasing the number of image layers. This hierarchical structure significantly reduces the computational effort of the network. The Feature Pyramid Networks (FPN) [58] in Figure 7 fit well with the multi-layer structure of the Swin Transformer and can resolve different-sized feature maps simultaneously. The Pyramid Pooling Module (PPM) [59] enhances the accuracy of the FPN by using multi-ratio global pooling in the output of the last layer of the Swin Transformer.

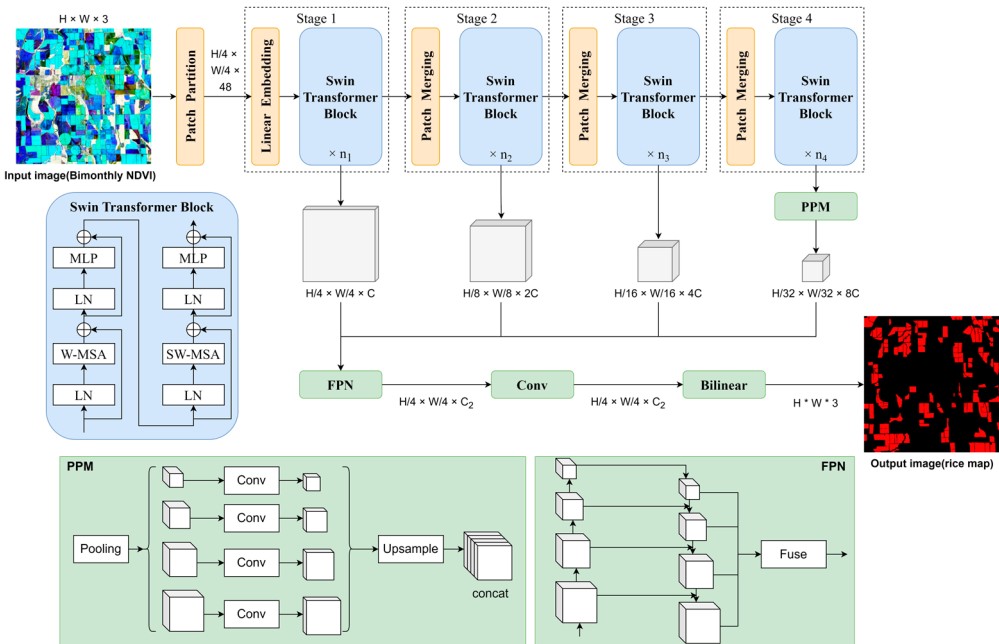

**Figure 7.** Swin Transformer for rice identification [56].

### 3.2. Evaluation Metrics

In this paper, model performance is evaluated by six metrics: mean intersection over union (*MIoU*), overall accuracy (*OA*), *recall*, *precision*, F1-score (*F1*) and *kappa*. These six metrics can be calculated using a confusion matrix, which is shown in Table 1. *MIoU*, *OA* and *kappa* are evaluated for the overall classification results (including rice and non-rice), and *recall*, *precision* and *F1* are evaluated for the rice classification results.

**Table 1.** Confusion Matrix (TP, FN, FP, and TN represent the number of pixels for each category, respectively. T and F refer to correct prediction and wrong prediction, respectively; P and N refer to the predicted results of rice and non-rice, respectively.).

| Confusion Matrix | | Prediction | |
|---|---|---|---|
| | | Rice | Non-Rice |
| **Ground Truth** | Rice | TP (True Positive) | FN (False Negative) |
| | Non-rice | FP (False Positive) | TN (True Negative) |

*MIoU* represents the intersection between the prediction and true values divided by the union. In the rice identification task, this can be used to reflect the degree of overlap between the prediction and reference data. The formula for representing *MIoU* in a binary classification task in terms of a confusion matrix is:

$$MIoU = \frac{TP + TN}{TP + 2FN + 2FP + TN} \tag{1}$$

*OA* represents the ratio of the number of correctly predicted pixels to the total number of pixels and can reflect the overall accuracy of the rice map predicted by the model. The calculation formula is expressed as:

$$OA = \frac{TP + TN}{TP + FN + FP + TN} \tag{2}$$

*Kappa* indicates the agreement between the predicted number of pixels and the true number of pixels for different categories and can be used to measure the balance of positive and negative samples. When the number of positive and negative samples is extremely unbalanced, the *kappa* value will be low even if the prediction accuracy is high. The calculation formula is expressed as:

$$Kappa = \frac{p_0 - p_e}{1 - p_e} \tag{3}$$

In the above equation, $p_0$ is equivalent to *OA*, and $p_e$ represents the sum of the multiplication of the true and predicted pixel counts for each class divided by the square of the total number of samples. In the binary classification, $p_e$ can be expressed as:

$$p_e = \frac{(TP + FP) \times (TP + FN) + (TN + FP) \times (TN + FN)}{(TP + FN + FP + TN)^2} \tag{4}$$

*Recall* indicates the proportion of correctly predicted rice pixels among all rice pixels in the reference rice map. *Precision* indicates the proportion of correctly predicted rice pixels among all rice pixels in the prediction rice map. *Recall* and *precision* are used to reflect the ability of the model to detect completely and accurately, respectively. The F1-score accounts for the balance of these competing metrics, which are usually incompatible. It balances *recall* and *precision* by calculating a harmonic mean of the two metrics. The F1-score is only high if both *recall* and *precision* are high. The calculation formulas of *recall*, *precision*, and *F1*-score are:

$$Recall = \frac{TP}{TP + FN} \tag{5}$$

$$Precision = \frac{TP}{TP + FP} \tag{6}$$

$$F1 = \frac{2 \times Recall \times Precision}{Recall + Precision} = \frac{2TP}{2TP + FN + FP} \tag{7}$$

### 3.3. Hardware and Training Settings

The three semantic segmentation networks were built based on PyTorch. The GPU of NVIDIA GeForce GTX 1080 with 8 GB of GDDR5X memory is used to accelerate training, and SDG (Stochastic Gradient Descent) optimizer with an initial learning rate of 0.01 is used during training the networks. Each network is trained for 200 epochs, and the models at the highest accuracy epochs are taken for comparative analysis.

## 4. Results

### 4.1. Rice Identification Accuracy

The evaluation results based on the comparison of the three semantic segmentation networks are shown in Table 2. From *OA*, *MIoU* and *kappa*, it can be seen that among the three semantic segmentation networks, the Swin Transformer extracts significantly better than DeepLab v3 and U-Net, and U-Net has the lowest accuracy. The Swin Transformer has 5.14%, 9.55%, 6.75%, 2.81% and 8.33% accuracy improvement in *OA*, *MIoU*, *precision*, F1-score and *kappa*, respectively, compared to DeepLab v3, and only *recall* is 1.03% lower than DeepLab v3. The Swin Transformer has 6.86%, 12.39%, 1.78%, 68.87%, 46.14% and 60.15% accuracy improvement over U-Net's *OA*, *MIoU*, *recall*, *precision*, F1-score and *kappa* six metrics, respectively.

**Table 2.** Model accuracy evaluation.

| Model | OA | MIoU | Recall | Precision | F1 | Kappa |
|---|---|---|---|---|---|---|
| U-Net [54] | 0.8934 | 0.8127 | 0.7652 | 0.4966 | 0.5526 | 0.4883 |
| DeepLab v3 [55] | 0.9080 | 0.8338 | **0.7868** | 0.7856 | 0.7855 | 0.7219 |
| Swin Transformer [56] | **0.9547** | **0.9134** | 0.7788 | **0.8386** | **0.8076** | **0.7820** |

Notes: the bold numbers indicate that the network has the highest accuracy value among the three networks.

The *precision* and *kappa* of U-Net are much lower than those of the other two models. U-Net is prone to misidentify non-rice as rice, and the agreement between the predicted and true values is low. The comparison between *precision* and *recall* shows that the Swin Transformer is more accurate for rice, while DeepLab v3 is more complete. The *F*1-score shows that the Swin Transformer is more accurate for rice in terms of the balance between accuracy and completeness.

### 4.2. Model Efficiency

The number of parameters, parameter sizes and time consumption during training and prediction for the three models are shown in Table 3. The number of parameters for U-Net is the lowest, approximately 1/5 of DeepLab v3. The Swin Transformer has the highest number of parameters, approximately 1.5 times that of DeepLab v3. Due to the difference in the number of model parameters, the prediction for a 1098 × 1098 bimonthly NDVI map also takes the least time for U-Net and the most time for the Swin Transformer. However, the Swin Transformer does not cause excessive training time due to the large number of parameters. For the training time, U-Net takes the shortest time, and DeepLab v3 takes the longest.

**Table 3.** Number of model parameters and time consumption.

| Model | Training Time (Per Epoch) | Prediction Time (Per Image) | Model Parameters | |
|---|---|---|---|---|
| | | | Amount (Million) | Size (MB) |
| U-Net [54] | 4 min 32 s | 3.60 s | 7.70 | 30.79 |
| DeepLab v3 [55] | 8 min 48 s | 3.90 s | 39.63 | 158.54 |
| Swin Transformer [56] | 7 min 27 s | 4.40 s | 62.31 | 249.23 |

Although the Swin Transformer has the longest prediction time, the difference with the other two networks is no more than 1 s. This shows that with GPU performance, the difference in model prediction time for a 1098 × 1098 3-band image is not significant enough to make a significant difference in practice, even with a large difference in the number of model parameters.

### 4.3. Prediction Results

Figure 8 represents the rice identification results of U-Net, DeepLab v3, and Swin Transformer in the test set. Group (b) shows the predicted values of non-rice pixels incorrectly identified as rice, corresponding to FP in the confusion matrix, with more incorrect pixels indicating lower *precision*. Group (c) shows the rice pixels not identified in the real values, which corresponds to FN in the confusion matrix, and more incorrect pixels indicate the lower *recall* of the model.

From the U-Net prediction results, it can be seen that there are many incomplete rice fields in the prediction images. U-Net very easily misidentifies non-rice as rice, which can reflect the same conclusion as U-Net's *precision* of only 0.4966 in Table 2.

The prediction results of DeepLab v3 can reflect that the rice fields identified by DeepLab v3 are more complete than those of U-Net, but the field boundaries are not accurate enough, and the predictions are shifted slightly to the northwest compared to the true values. Compared to U-Net and the Swin Transformer, fewer rice pixels incorrectly identified non-rice in DeepLab v3's results. This indicates that DeepLab v3 has a more

complete identification of rice, which corresponds to the highest *recall* of DeepLab v3 in Table 2.

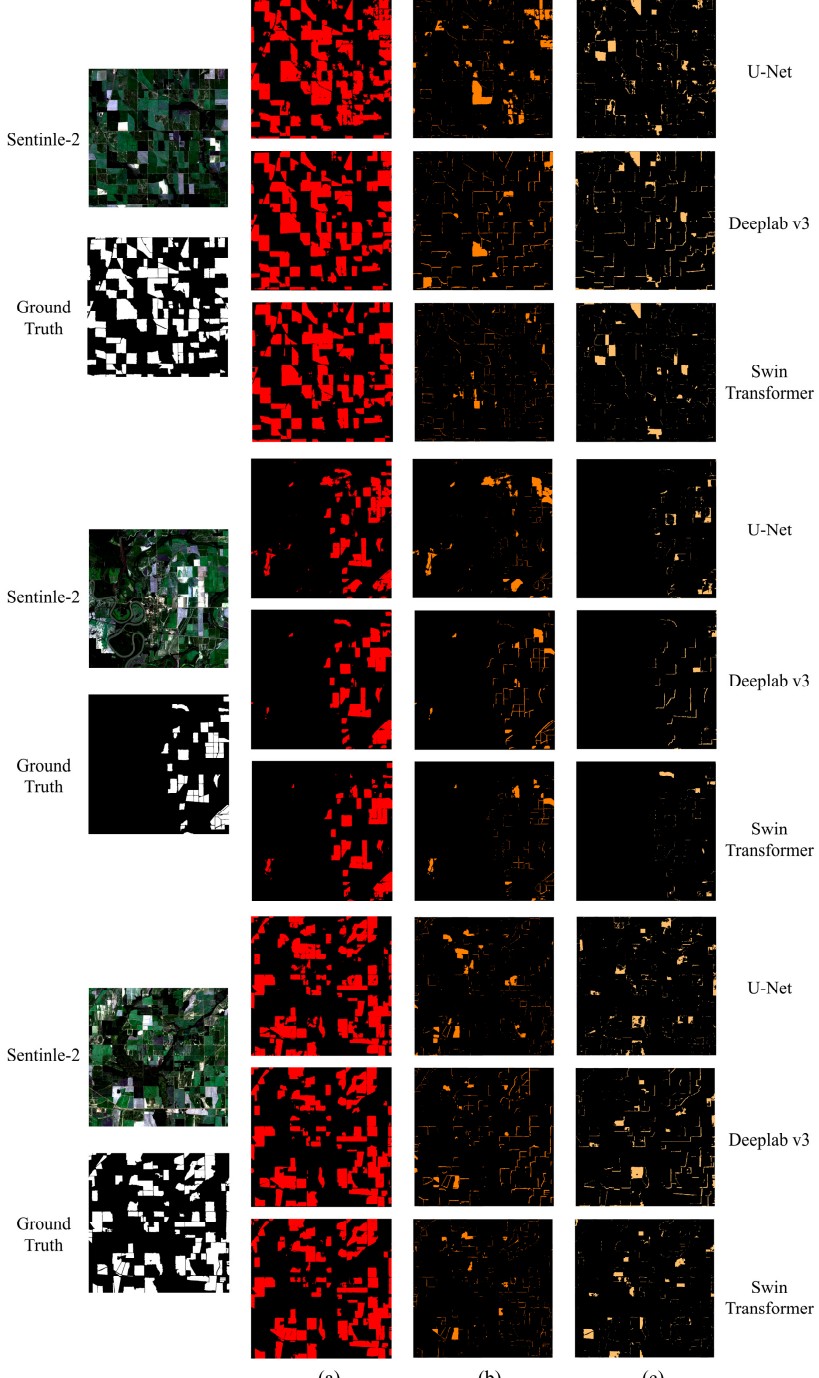

**Figure 8.** Rice identification prediction results. (**a**) Rice prediction map. (**b**) Non-rice wrong prediction map. (**c**) Rice warning prediction map.

The Swin Transformer's prediction results are more accurate than those of U-Net and DeepLab v3. The Swin Transformer identifies complete rice fields with accurate boundaries. The results for the Swin Transformer in group (b,c) show that the rice field boundaries identified by the Swin Transformer differ from the true values by only approximately one pixel. Compared to U-Net and DeepLab v3, fewer non-rice pixels were incorrectly identified as rice in the Swin Transformer's results. This indicates that the Swin Transformer

has a more accurate recognition of rice, which corresponds to the highest *precision* of the Swin Transformer in Table 2.

Details of rice identification by the three semantic segmentation networks are shown in Figure 9. All three networks show varying degrees of incomplete recognition in the areas circled by the blue circles in group (a), with U-Net missing the most rice pixels and the Swin Transformer missing the fewest. In group (b), where the yellow circles are circled, U-Net appears to misidentify non-rice as rice over a large area and, relatively, DeepLab v3 and the Swin Transformer have no prediction errors here. Purple circles in group (c) show the identification of the three networks at the rice field boundaries. Both U-Net and the Swin Transformer were able to obtain accurate field boundaries, but DeepLab v3 identified the field boundaries less accurately. For some areas, U-Net identified the field boundaries even more accurately than the Swin Transformer.

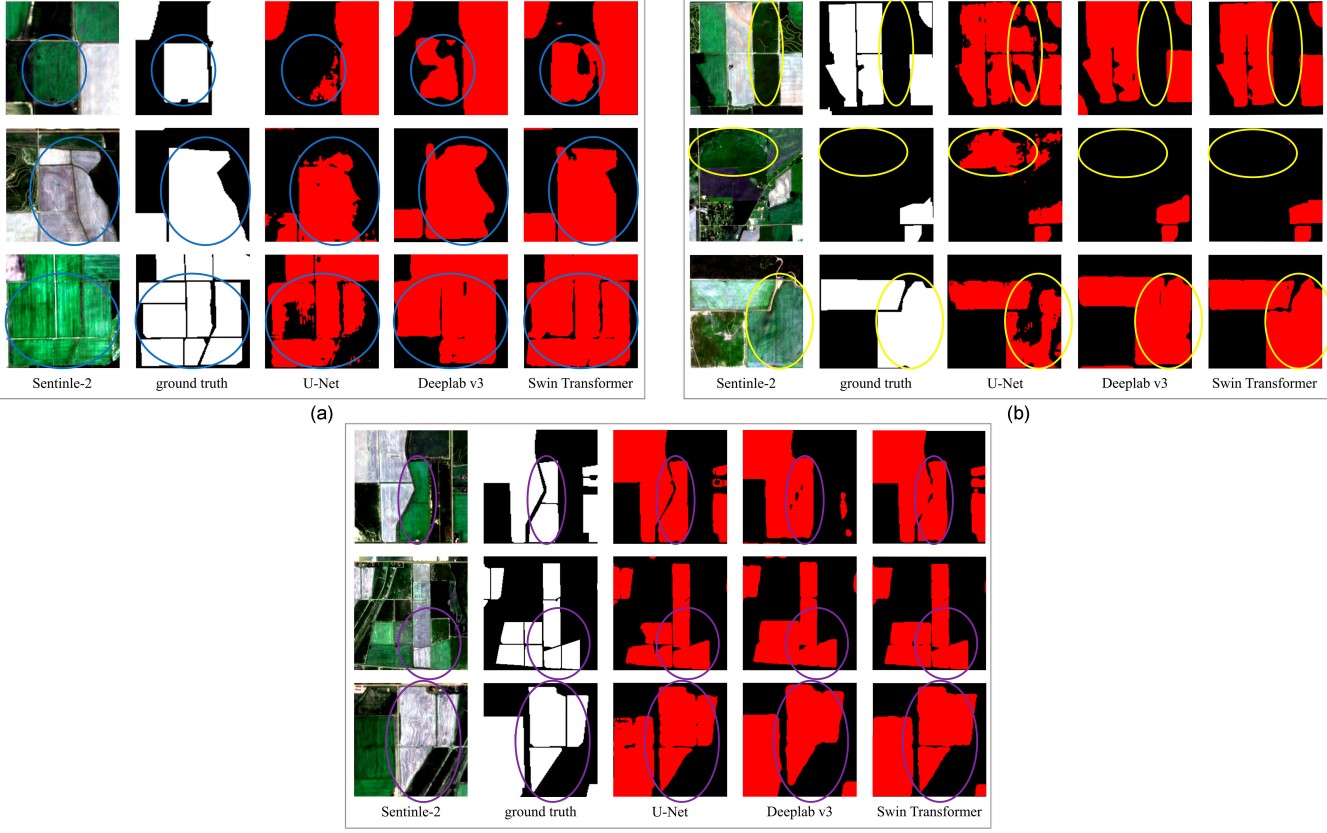

**Figure 9.** Details of the prediction results. (**a**) Incomplete rice field. (**b**) Large misidentified area. (**c**) Field boundaries in the prediction.

### 4.4. Misclassified Crops

The percentages of land use types in the test data are shown in Figure 10, and the crops include rice, soybeans, corn and cotton. Figure 10 shows that soybean, woody wetlands, rice, forest and fallow are the most dominant types of land use in the test data.

The total number of pixels that the three networks, U-Net, DeepLab v3 and Swin Transformer, incorrectly identified non-rice as rice (noted as "non-rice misidentification") is 940,626, 409,713 and 297,839, respectively, as shown in Figure 11. Figure 11 also shows the number of the misidentified pixels in terms of the land use types. The land use types being mostly misidentified by the three networks are soybean, followed by corn. Fallow, forest and woody wetland, which are accounted for a large proportion of the test data, are less misidentified by all the three networks.

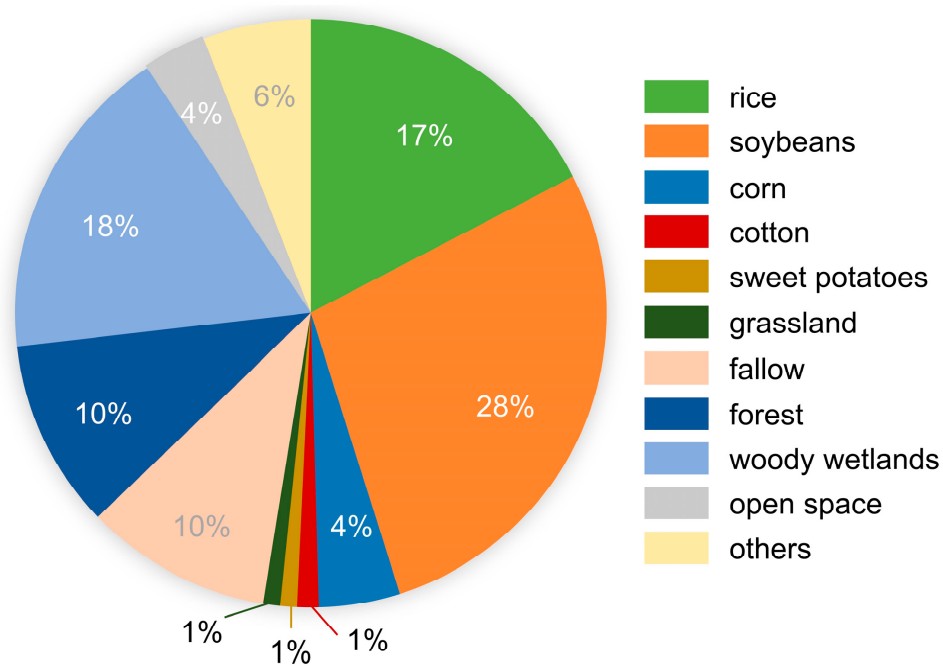

**Figure 10.** Percentage of each land use type in the prediction images.

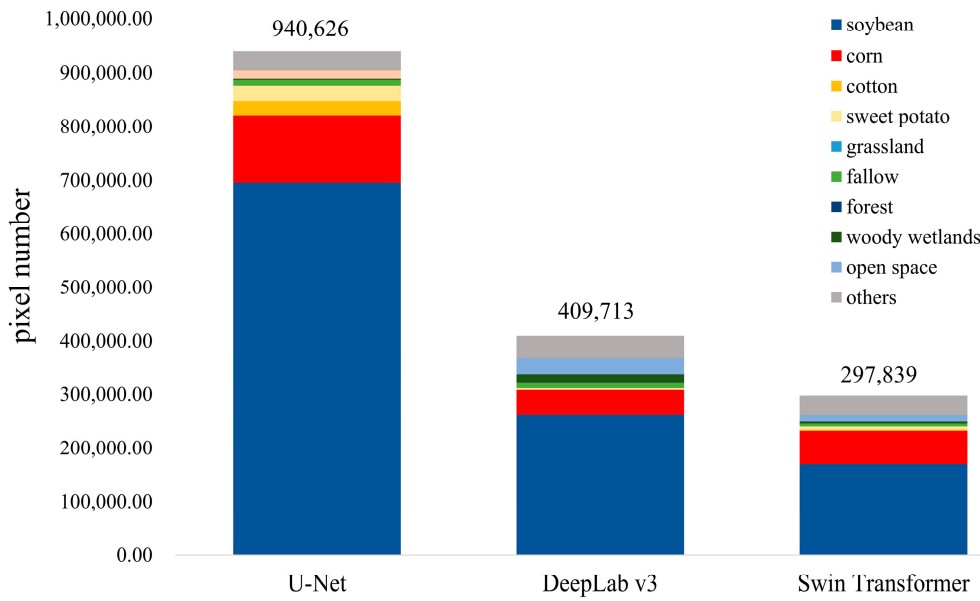

**Figure 11.** The number of the misidentified pixels in terms of the land use types.

The misidentified proportion for each land use type is shown in Table 4. It is calculated with Equation (8). Table 4 shows that the Swin Transformer generally has better performance than the U-Net and DeepLab v3 in the non-rice misidentification issue. The U-Net misidentified crops (sweet potato, corn, soybean and cotton) mostly and the DeepLab v3 misidentified non-crops (fallow, forest, woody wetland, open space, others) mostly.

$$P = \frac{FP_i}{Num_i} \qquad (8)$$

where $i$ be the land use type, $FP_i$ be the number of pixels being misidentified as rice for type $i$, and $Num_i$ be the total number of pixels in the test data for type $i$.

**Table 4.** The proportion of "non-rice misidentification" for each land use type.

|  | Soybean | Corn | Cotton | Sweet Potato | Grassland | Fallow | Forest | Woody Wetland | Open Space | Others |
|---|---|---|---|---|---|---|---|---|---|---|
| U-Net [54] | **20.75%** | **23.39%** | **18.41%** | **26.85%** | 0.02% | 0.89% | 0.002% | 0.09% | 3.60% | 5.07% |
| DeepLab v3 [55] | 7.84% | 8.44% | 0.37% | 2.68% | **0.11%** | **0.91%** | **0.03%** | **0.70%** | **7.06%** | **5.94%** |
| Swin Transformer [56] | 5.08% | 11.52% | 1.41% | 5.43% | 0.03% | 0.43% | 0.002% | 0.17% | 2.93% | 5.16% |

Notes: the bold numbers indicate that the network has the most misidentified pixels among the three networks.

## 5. Discussion

We evaluate and compare the rice identification performance of three semantic segmentation networks in terms of rice identification accuracy, model efficiency and the agreement of predictions with ground truth. At the same time, we analyzed the land use types that are easily confused with rice in the rice identification results of the three networks.

### 5.1. Analysis of Rice Identification Results

The best performing of the three networks is Swin Transformer, which has the highest rice identification accuracy and the boundaries of the identified rice fields matches well with the ground truth. The next best network is DeepLab v3, which also has a high accuracy, but the identified rice fields are shifted to the upper left corner from the ground truth. U-Net's rice identification accuracy is the lowest but can obtain good boundary segmentation details for correctly identified rice fields and in some places even outperform Swin Transformer.

The rice identification results of U-Net is due to the shallow depth and lightweight structure of U-Net, which leads to incomplete learning of rice features in a complex rice identification task with a small quantity of data in this paper. However, the segmented upsampling and addition of convolutional layers in the upsampling phase of U-Net enable it to restore the rice field boundary information very well.

DeepLab v3 benefits from its parallel atrous convolution structure, which allows it to obtain highly accurate rice identification results. However, its eight-fold reduced feature maps are upsampled directly to the original resolution by bilinear interpolation, which can result in less accurate boundaries. The reason for the shift to the upper left in the DeepLab v3 results is that the main method for reducing the image resolution is by convolution with a stride of 2. When the stride is 2, for the case where the image resolution is even and the convolution kernel is odd, there is a missing row at the bottom right, as shown in Figure 12. When this accumulates through multiple layers, it causes the positive features in the feature map to gradually shift toward the upper left corner [60], which causes the rice fields identified by DeepLab v3 to be shifted to the upper left.

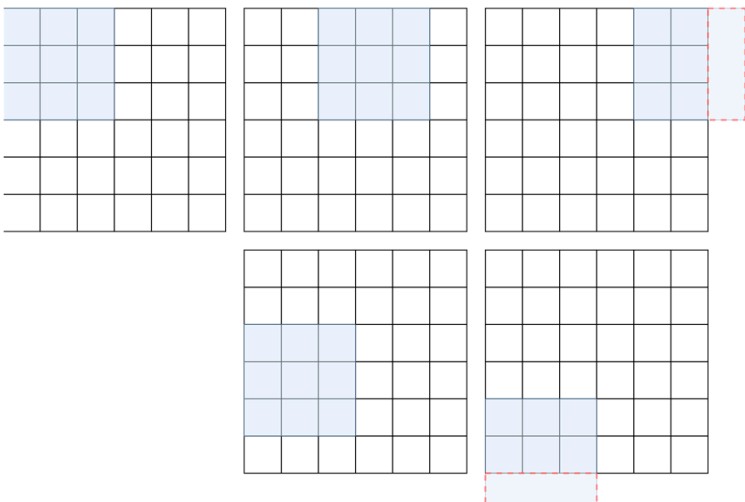

**Figure 12.** Schematic of convolution with stride = 2.

Swin Transformer combines the advantages of U-Net and DeepLab v3 to obtain both highly accurate and highly detailed rice identification results. This is because the attention module in the Swin Transformer calculates the relationship between all pixels in a window to obtain the most deserving regions in the image, which allows it to have a stronger global feature extraction capability than CNN. Because Swin Transformer has excellent ability to extract both local and global features at the same time, it can not only distinguish the color, shape, texture and other local information of rice and non-rice, but also obtain the relative position information between rice fields and non-rice fields. This enables Swin Transformer to obtain high-precision, high-detail rice extraction results.

### 5.2. Analysis of Model Efficiency

The predict time for the three networks is not significantly different, with each image taking less than 5 s. U-Net has the lowest number of model parameters and the shortest training time, while Swin Transformer has the highest number of model parameters but the training time for the model is less than DeepLab v3.

The reason that the DeepLab v3 training time is longer than that of the Swin Transformer is that the training time of the model is not simply related to the number of parameters of the model but rather to the computational complexity of the model. For each layer, the computational complexity of convolutional and self-attention is shown in the following equation [44]:

$$C(Self - Attention) = O\left(n^2 d\right) \tag{9}$$

$$C(Convolutional) = O(knd) \tag{10}$$

where $n$ is the number of pixels, $d$ is the image dimension, and $k$ is the convolution kernel size. In DeepLab v3, as the model becomes deeper, the dimensionality of the feature map increases significantly to a maximum of 2048. In contrast, in the Swin Transformer, the feature map dimensionality is related to the embedding dimension, which, in this experiment is a maximum of 768. Therefore, in the experiments in this paper, the training time consumed by DeepLab v3 is longer than that of the Swin Transformer.

It can be seen that the Swin Transformer has great advantages in terms of accuracy and efficiency. Therefore, it has great application potential for large-scale (country scale or even global scale) crop identification.

### 5.3. Analysis of Misidentification

In general, the Swin Transformer has also better performance than the U-Net and the DeepLab v3 in reducing the non-rice misidentification. This confirms that the context information globally obtained by the self-attention structure of Swin Transformer is effective to reduce the non-rice misidentification.

Compared with the crops, the non-crops are much less misidentified as rice since the time-series NDVI feature play an important role on distinguishing crops with the non-crops. Among the misidentified four crops, we also found that the soybean is misidentified mostly among them. This could be related to the fact that the soybean is the most abundant crop in the study area, thereby increasing the probability of misidentification.

In crops, the three networks tend to misidentify soybean, corn, sweet potatoes and cotton as rice. This is because the four crops are similar to rice in that they are summer crops planted in April and harvested from late September to October and show a similar pattern in the bimonthly NDVI map. The NDVI changes for these five crops in the study area are shown in Figure 13. The NDVI values of soybean, corn, sweet potato and cotton in the study area possess a similar trend of flat growth followed by a rapid rise and then fluctuating decline to that of rice, which has led to these four crops being easily misidentified as rice.

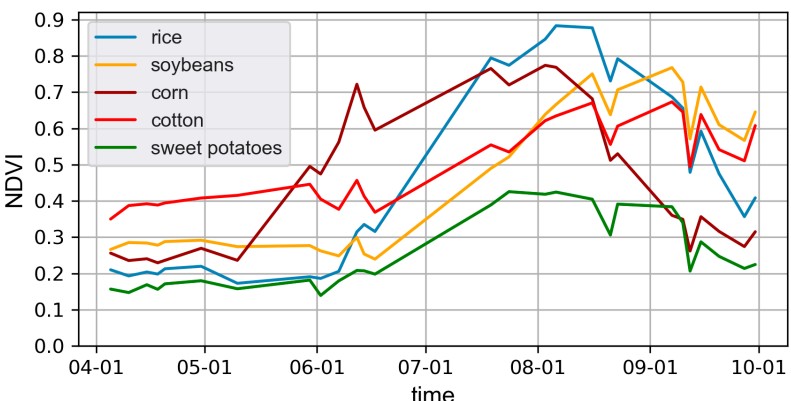

**Figure 13.** NDVI values for different classes of pixels over 1 April 2020–30 September2020.

## 6. Conclusions

In this study, we explore the performance of three rice identification networks based on Sentinel-2 images. All three networks are semantic segmentation networks, including two CNN networks (U-Net and DeepLab v3) and one Vision Transformer network (Swin Transformer).

The study shows that Swin Transformer has the highest accuracy in the rice identification task based on 10 m resolution Sentinel-2 and can segment the rice field boundary information very accurately. Although Swin Transformer has the highest number of model parameters, the training time is less than DeepLab v3. This indicates that Swin Transformer is a semantic segmentation network with high accuracy, precise details and high training efficiency.

The two CNN networks have their own advantages and disadvantages for the rice identification task in this paper. DeepLab v3 has high accuracy and can recognize most of the rice fields, but the recognition results tend to be shifted to the upper left corner by about two pixels relative to the ground true, and the network takes the longest time to train. The U-Net is a lightweight network, with a much smaller number of model parameters and training time than the other two networks and can segment very fine field boundaries. However, due to the small number of layers and low complexity of the U-Net network, the accuracy of rice identification is low for the small amount of training data and complex background information in this paper, and it is easily confused with soybean, maize, cotton and sweet potato, which are also summer crops.

This study compares the performance of three semantic segmentation networks in rice identification. The results show the great potential of the attention-based mechanism of the Vision Transformer network for recognizing crops from meter resolution remote sensing images. Based on this study, we will further design a model architecture that combines the advantages of CNN and Transformer to extract rice information from Sentinel-2 images. And we will investigate the characteristics and advantages of more deep learning models with different mechanisms to explore better crop identification networks.

**Author Contributions:** H.X.: Conceptualization, methodology, software, validation, formal analysis, data curation, writing—original draft preparation; J.S.: conceptualization, methodology, software, writing—review and editing, project administration, funding acquisition, supervision; Y.Z.: writing—review and editing, supervision. All authors have read and agreed to the published version of the manuscript.

**Funding:** This research was funded by the National Key Research and Development Program of China (2021YFE0117800, 2022YFF0711602), the 14th Five-year Informatization Plan of Chinese Academy of Sciences (CAS-WX2021SF-0106), and the National Data Sharing Infrastructure of Earth System Science (http://www.geodata.cn/, accessed on 11 January 2021).

**Data Availability Statement:** Publicly available datasets were analyzed in this study. The 2020 Cropland Data Layer dataset is available following this link: https://nassgeodata.gmu.edu/CropScape/, accessed on 11 January 2021.

**Acknowledgments:** We would like to acknowledge the United States Department of Agriculture for providing the national agricultural statistics service.

**Conflicts of Interest:** The authors declare no conflict of interest.

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
