# Peer review of "Evaluation and Comparison of Semantic Segmentation Networks for Rice Identification Based on Sentinel-2 Imagery"

_remotesensing, doi:10.3390/rs15061499_

Round 1

Reviewer 1 Report

Dear Author,

I have reviewed your manuscript and found it to be well-structured and easy to read. The content provides valuable insights for the field of remote sensing engineering. However, while the paper is informative, the level of innovation could be improved. I suggest two possible ways to enhance the paper:

Firstly, I recommend designing a model architecture that incorporates the strengths of both CNN and Transformer to extract rice information from Sentinel-2 imagery. This can significantly enhance the accuracy of rice extraction from satellite images. It would be beneficial to provide a detailed description of the model architecture and its implementation.

Secondly, I suggest that you elaborate on the mechanism behind the model performance in the discussion section. For example, it would be helpful to compare the global feature extraction capability of Transformer-based models to CNN-based models, and highlight how the model was adapted to the experimental scenarios.

Reviewer 2 Report

Authors compare three semantic segmentation networks in terms of rice identification accuracy and model efficiency.

The paper is well written, results are convincing and a good discussion is also performed at the end of the results.

In my opinion the paper is worth of publication in this form.

Author Response

Thanks. We appreciate your positive comments.

Reviewer 3 Report

The study compares the performance of three methodologies based on the semantic segmentation networks for the identification of rice crops using Sentinel-2 data.

The topic is of high interest and important results have been achieved and discussed.

I suggest only few integrations:

-       Add the references of the maximum composite method in the §2.2.1 to clear better the proposed methodology;

-       Please define in the text in the § 3.2 the meaning of TP, FP, FN, and FP;

-       Discuss better the comparison with the land use data in the §4.4; it is not clear how these data are compared with the rice identification. 
